# Extreme Wave Analysis for the Dubai Coast

Khaled Elkersh [1] , Serter Atabay [1,*] and Abdullah Gokhan Yilmaz [2]

1 Department of Civil Engineering, American University of Sharjah,
  Sharjah P.O. Box 27272, United Arab Emirates
2 Department of Engineering, La Trobe University, Melbourne, VIC 3086, Australia
* Correspondence: satabay@aus.edu

**Abstract:** This paper aims to present the result of commonly used extreme wave analysis distribution methods applied to a long-term wave hindcast at a point in the Arabian Gulf near the coastline of Dubai, United Arab Emirates. The wave data were hindcasted for a total period of 40 years, starting from 1 January 1979 to 31 December 2018. This analysis aims to support the design, repair, and maintenance of coastal structures near the Dubai coast. A 2.5 m threshold is selected using the Peak Over Threshold method to filter the storm data for the extreme wave analysis. Different distribution methods are used for this analysis such as Log-normal, Gumbel, Weibull, Exponential, and Generalized Pareto Distribution (GPD). The significant wave heights are predicted for different return periods. The GPD distribution appears to fit the data best compared to the other distribution methods. Many coastal projects are being planned near the Dubai coastline. Hence, the analysis presented in this paper would be useful in designing safe and efficiently designed projects.

**Keywords:** climate change; extreme wave analysis; wave height; return period

## 1. Introduction

Recently, coastal areas around the world have been under significant marine and coastal development [1]. In particular, the coastline of Dubai, United Arab Emirates has experienced significant offshore and onshore landscape development over the past years [2]. Designing different types of marine and coastal structures requires accurate knowledge and analysis of long-term data about different environmental conditions such as waves. For instance, the design of safe and economic coastal and offshore structures is based on the design wave heights for different return periods [3]. Similarly, the safety of existing structures is affected by climate change as it influences extreme wave heights [1]. Therefore, the proper quantification of wave statistics, such as the extreme wave height, is a critical part of coastal structure design. For example, the poor estimation of the design wave height of a structure could result in a poor design that could lead to structural failure or expensive overdesign [4,5].

Climate change is one of the challenges that affects cities, coasts, agriculture, water resources, and natural ecosystems [6,7] all over the world [8,9]. However, coastal zones are very vulnerable to its impacts. Coasts and beaches suffer from coastal processes such as sea-level rise, coastal flooding and erosion, and storm surge, which substantially damage the coastal infrastructure [10–14]. Chini et al. (2010) performed extreme event analysis to estimate climate change implications on inshore waves and the occurrence of extreme events [7]. The study showed that wave statistics are sensitive to sea-level rise. Additionally, the authors suggest that climate change leads to a significant increase in extreme wave heights and the frequency of occurrence of extreme waves [7].

Poor knowledge of the nature of the wave climate can increase the risk associated with a coastal or marine project [15]. Neelamani et al. (2007) suggest that many coastal structures in the Arabian Gulf appear to be overdesigned since there is not sufficient extreme wave analysis work done for different return periods [16]. Generally, extreme

wave analysis provides a theoretical distribution of the probability of occurrence of different wave parameters over a long period of time [17,18]. Various studies regarding extreme wave analysis have been conducted to date. Mathiesen et al. (1994) argue that using statistical analysis of extreme waves for selecting a proper design wave height is central in coastal engineering [19]. The authors also present recommended methods and practices for statistical analysis of extreme significant wave heights. Similarly, Goda (2010) provides an in-depth review on this topic [20]. This paper uses extreme wave analysis distributions to develop relationships between large wave heights and their corresponding reduced variates. These relationships are then used to extrapolate significant wave heights at different return periods.

In this study, a long-term wave hindcast record of 40 years (1979–2018) is analyzed to estimate accurate significant wave heights for different return periods by fitting different statistical distributions. Numerical wave hindcasts are often used for extreme wave analysis due to the lack of long-term observational wave records [21,22]. Furthermore, threshold selection is a critical part of extreme wave analysis [23]. The Peak Over Threshold (POT) method is implemented to extract statistically independent storm peaks above a specific threshold. The distribution methods used for this analysis are conventional methods such as Log-normal, Gumbel, Weibull, and Exponential distributions. In addition to these conventional methods, the Generalized Pareto Distribution (GPD) will be used as it is considered a good-performing candidate distribution [4,24]. These methods are all used and compared as there is not a single distribution method that could fit all wave datasets. Therefore, the selected method is the one that provides the best fit for the wave data. The least squares method presented by Goda (2010) is selected among the several available fitting methods for this study [20]. In this study, the best-fitting distribution is identified using the value of the correlation coefficient (r) between the ordered data and their reduced variates, which is a test for goodness of fit. The extreme values are extracted for different return periods, such as 1, 20, 50, 100, and 200 years.

The study area of this paper is a point in the Arabian Gulf near the coastline of Dubai, United Arab Emirates. The Arabian Gulf is a strategically important and very active marine area that includes the largest offshore oil and gas fields in the world [25]. The Arabian Gulf is an extension of the Indian Ocean through the Strait of Hormuz. The total area of the gulf is approximately 226,000 km$^2$. It is 990 km long and varies in width from 56 to 338 km with an average depth of about 35 m, separating Iran from the Arabian Peninsula. The Arabian Gulf is located between the longitude of 48°–57° East and the latitude of 24°–30° North [3]. In the UAE's territorial waters in the Arabian Gulf, the water depths extend to a maximum of 50 m [26]. The data point is at a depth of 15 m below the Dubai municipality Datum and is located about 2.5 km from the Palm Jumeirah Island, offshore the Dubai coastline (25°09′00.0″ N, 55°06′00.0″ E), as shown in Figure 1.

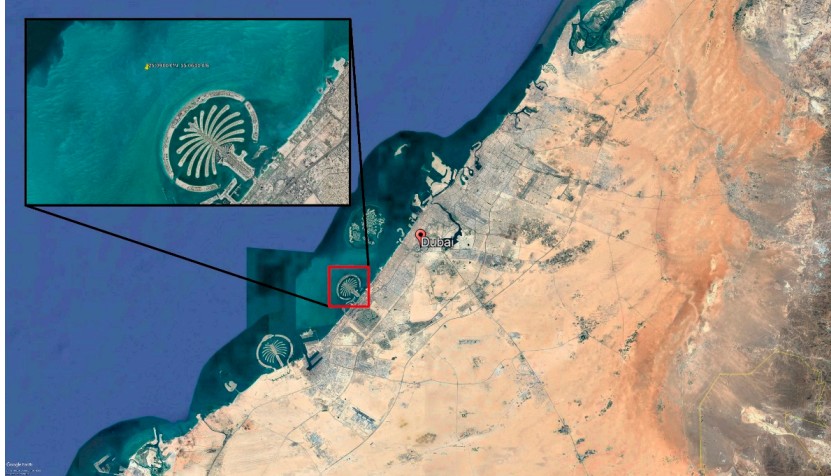

**Figure 1.** Location of the study point (25°09′00.0″ N, 55°06′00.0″ E).

## 2. Materials and Methods

The wave data in this analysis are a hindcast of 40 years of hourly data computed numerically starting from the 1 January 1979 to 31 December 2018 [26]. The numerical model is forced with Climate Forecast System Reanalysis (CFSR) wind fields from the National Centers for Environmental Prediction (NCEP) reanalysis dataset, which is available from January 1979 to 2010 [27]. The hindcast is developed using a SWAN wave model, which is validated against a set of field measurements and satellite altimeter data. This hindcast is composed of hourly spectral wave data that specify significant wave height, peak wave period, and peak wave direction. In general, the northwest direction is the most dominant wave direction in the Arabian Gulf [16,26]. Therefore, the wave data are filtered by wave direction to consider only the waves coming from the most critical direction (northwest) to be used in the extreme wave analysis, which makes the analysis more specific. On the other hand, by considering all directions, minor or no changes might be seen in the results as the waves coming from the other directions are much smaller compared to the northwest direction.

The choice of the candidate probability distribution methods is an empirical step before starting the extreme wave analysis of the data. Several extreme value distribution methods could be used for extreme wave analysis, for example, Gumbel, Fréchet, FT-III, Lognormal, Weibull, Exponential, GPD, and others. However, some of them might not be suitable for this study. For instance, FT-III might not be a suitable distribution method as it converges quite slowly [28]. Similarly, some authors seem to argue that the Fréchet distribution tends to overestimate wave heights for long return periods [29]. Therefore, the distribution methods adopted in this study are Log-normal, Gumbel, Weibull, Exponential, and GPD distributions.

The Peak Over Threshold method is a commonly accepted method for the extraction of storms or extreme waves [30]. The basic definition of a storm can be the time when wave height exceeds a certain threshold [1]. Therefore, a threshold is selected first, and all unique storms above that threshold are filtered and represented by their corresponding peak wave heights. However, deciding on a threshold using the POT method could be a subjective and difficult decision. A high threshold could produce too few storm events and a low threshold could produce too many. In this case, a 2.5 m threshold is selected, which yields 113 independent storms in the span of 40 years. This selected threshold corresponds to the 99th percentile of the used wave data. To ensure that storms are independent, two consecutive storms must be at least one day apart; otherwise, the two storms are counted as a single storm. The obtained storms are ranked in descending order of peak wave heights to be further analyzed. Figure 2 shows a plot of the significant wave heights of the ordered 113 storm events. Each isolated storm is represented by its peak wave height.

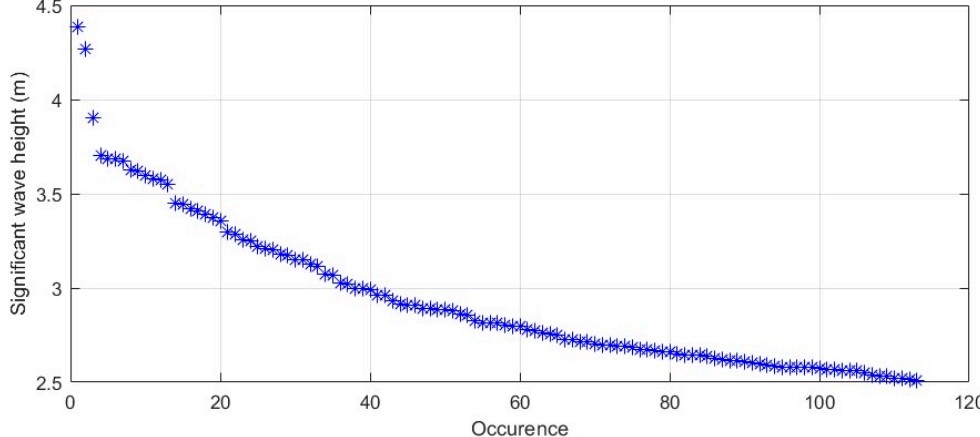

**Figure 2.** A plot of the peak wave heights of the 113 storm events at the Dubai coast coming from the northwest direction in the period 1979–2018.

The plotting position formula is used to calculate the probability of exceedance ($Q$) and the probability of non-exceedance ($P$) of the wave heights of the ordered storms. Several plotting position formulas are available for unbiased estimation of the return value. The unbiased plotting position formula that satisfies the different distribution functions is shown in Equation (1)

$$Q = \frac{i - c_1}{N + c_2}; \; P = 1 - Q \tag{1}$$

where (i) and (N) are the ranking of the data points and the total number of points, respectively. Additionally, $c_1$ and $c_2$ are constants for unbiased plotting position for each distribution shown in Table 1. For the Weibull and GPD distributions, the constants are a function of ($\alpha$), which is the shape parameter for the function [17,28]. Tyralisa et al. (2019) suggest that the significance of the shape parameters comes from the fact that it determines the behavior or the shape of the upper tail of the distribution [31]. The higher the value of the shape parameter, the heavier is the tail of the distribution. In this case, the value for $\alpha$ is determined by trial and error, since it influences both the plotting position and the curvature of the graph.

**Table 1.** Constants of unbiased position plotting.

| Distribution | $c_1$ | $c_2$ |
| --- | --- | --- |
| Log-Normal | 0.25 | 0.125 |
| Gumbel | 0.44 | 0.12 |
| Exponential | 0.47 | 0.43 |
| Weibull | $0.20 + 0.27/\sqrt{\alpha}$ | $0.20 + 0.23/\sqrt{\alpha}$ |
| GPD | 0.45 | 0 |

The general formulas for the distribution methods are provided in the equations below [1]. The least squares method is used by transforming these distribution functions into a simple linear form and then fitting the linear equation shown below to the wave data to estimate the distribution parameters [20,30].

$$Y = AX + B \tag{2}$$

where $Y$ and $X$ are the transformed probability axis and the transformed wave height axis, respectively. The coefficients $A$ and $B$ are the slope and intercept of the straight-line relationship, respectively.

The Log-Normal, Gumbel, Weibull, Exponential, and GPD distributions are candidate distributions for the extreme value analysis of the ordered data. The Log-normal distribution is one of the distributions that were first used for extreme-value analysis [28]

$$P = \Phi\left(\frac{lnH - \overline{lnH}}{S_{lnH}}\right) \tag{3}$$

The equation is transformed to be

$$Y = \Phi^{-1}(P); X = lnH; A = \frac{1}{S_{lnH}}; B = -\frac{\overline{lnH}}{S_{lnH}} \tag{4}$$

In addition to Log-normal distribution, the Gumball distribution is developed specifically for the analysis of extreme values shown in Equation (5)

$$P = exp\left(-exp\left(-\frac{H - \gamma}{\beta}\right)\right) \tag{5}$$

The equation is transformed to be

$$Y = -ln\left(ln\frac{1}{P}\right) = G; X = H; A = \frac{1}{\beta}; B = -\frac{\gamma}{\beta} \tag{6}$$

where $\beta$ and $\gamma$ are the scale and location parameters, respectively. The Weibull distribution is a three-parameter distribution function shown in Equation (6) and is the most used distribution method for extreme wave analysis [28].

$$P = 1 - exp\left(-\left(\frac{H-\gamma}{\beta}\right)^{\alpha}\right) \tag{7}$$

The equation is transformed to be

$$Y = \left(ln\frac{1}{Q}\right)^{\frac{1}{\alpha}}; X = H; A = \frac{1}{\beta}; B = -\frac{\gamma}{\beta} \tag{8}$$

In addition to the scale and location parameters, the Weibull distribution has a third parameter, which is the shape parameter ($\alpha$). The Exponential distribution shown in Equation (9) is a candidate distribution method. It is a reduced version of the Weibull distribution when $\alpha$ is equal to 1 [28].

$$P = 1 - exp\left(-\left(\frac{H-\gamma}{\beta}\right)\right) \tag{9}$$

The equation is transformed to be

$$Y = \left(ln\frac{1}{Q}\right); X = H; A = \frac{1}{\beta}; B = -\frac{\gamma}{\beta} \tag{10}$$

Sulis et al. (2017) suggest that the Generalized Pareto Distribution is one of the most performing credible candidate distributions for extreme wave analysis [24]. It is a three-parameter function that works in two forms depending on the shape parameter $\alpha$ as shown in Equation (11) [32].

$$P = \begin{cases} 1 - \left(1 - \alpha\left(\frac{H-\gamma}{\beta}\right)\right)^{\alpha} & \alpha \neq 0 \\ 1 - exp\left(-\left(\frac{H-\gamma}{\beta}\right)\right) & \alpha = 0 \end{cases} \tag{11}$$

When $\alpha = 0$, the GPD distribution is said to correspond to the Exponential distribution and is linearly transformed as shown in Equation (10). When $\alpha \neq 0$, the GPD function is transformed to

$$Y = \frac{1 - (Q)^{\alpha}}{\alpha}; X = H; A = \frac{1}{\beta}; B = -\frac{\gamma}{\beta} \tag{12}$$

Extreme waves are defined in terms of their wave heights and return periods. The return period is the average recurring interval between successive storms at a given wave height [28]. The wave heights for different return periods are extrapolated from the different distribution models created. The exceedance probability of an event could be estimated using Equation (13)

$$Q = \frac{1}{\lambda T_R} \tag{13}$$

where $\lambda$ is the number of events per year on which the analysis is based, and $T_R$ is the return period. The exceedance probability is then used with the general formula for each distribution to extrapolate the wave heights.

The Log-normal, Gumbel, Weibull, Exponential, and GPD distributions could all be a good fit for the storm data. However, the best-fit distribution should meet the goodness-of-fit selection criteria [20,28]. You and Yin (2013) argue that the goodness of fit criterion could

be used to select the best-fit distribution function between several candidate distributions used on the same wave data [28]. An indicator for this criterion is the correlation coefficient (r) or the sum of squares of the error ($R^2$). The closer either of these values to 1, the better this distribution fits the data points. Therefore, the best-fit distribution should be the adopted distribution for further wave height analysis and design.

## 3. Results

Five extreme wave analysis distribution methods are compared for the goodness to fit the storm data. The Weibull and GPD distributions have a third parameter in their distribution functions that needs to be determined, which is the shape parameter ($\alpha$). For these distributions, the value for $\alpha$ is determined by trial and error. For the Weibull distribution, the value of the shape parameter is restricted between 1 and 2 for extreme wave analysis as suggested by You (2013) [28]. The shape parameter for the GPD function is tested for values between $-1$ and 1. Figure 3 shows the value of $\alpha$ that yields different correlation coefficient r for each of the candidate distribution functions. In particular, $\alpha = 1.2$ gives the highest correlation coefficient for the Weibull distribution function. Similarly, the shape parameter for the GPD function with the highest correlation coefficient is $\alpha = 0.2$.

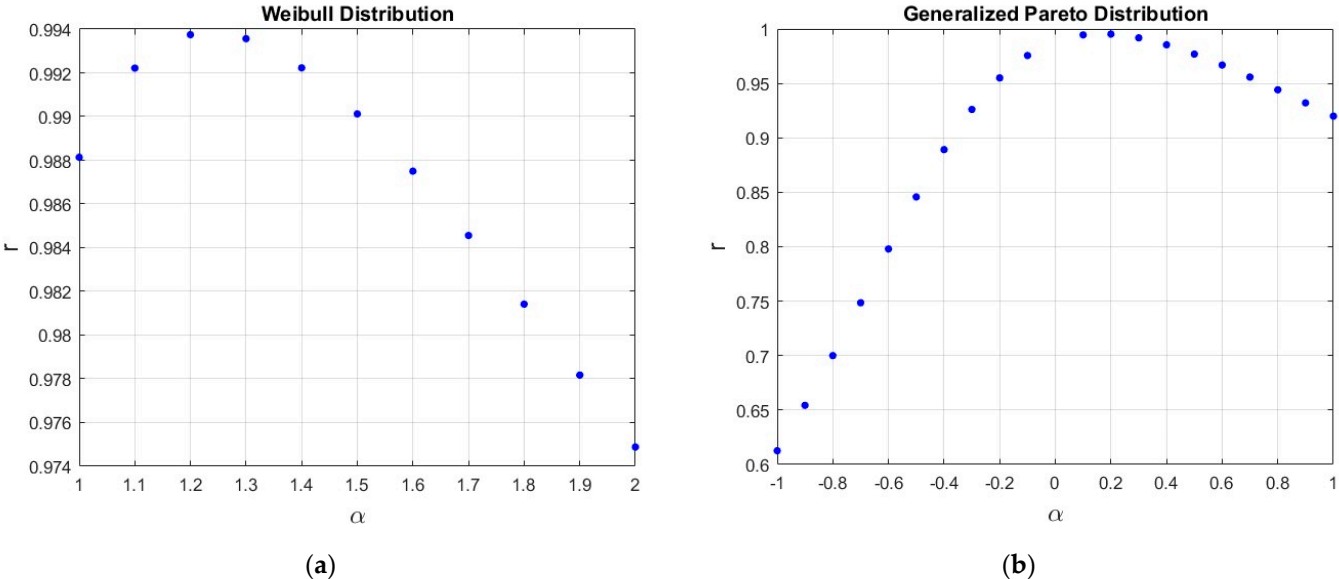

(**a**)                                                                    (**b**)

**Figure 3.** Correlation coefficient (r) values for different shape parameter values ($\alpha$) for the selected candidate distribution functions. The maximum value of r gives the value of $\alpha$ of the candidate function that best fits the wave dataset: (**a**) Weibull distribution; (**b**) Generalized Pareto Distribution (GPD).

The ordered 113 peak storms above the 2.5 m threshold are extracted, and their probability of exceedance is calculated. By using the probability of exceedance for each storm, the reduced variates are calculated for each distribution. Table 2 shows sample calculations for the first 12 peak storms for the distribution methods. The variables titled Z, G, W, E, and U correspond to the reduced variates for the Log-normal, Gumbel, Weibull, Exponential, and GPD distributions, respectively. Different plots are generated for the different extreme wave analysis methods used. Figure 4 shows the different plots generated for peak wave heights for each method. The plots represent the relationship between the wave heights of the peak storms against the reduced variates. The figure also shows the equation of the best fit and the sum of squares of the error ($R^2$) for each distribution.

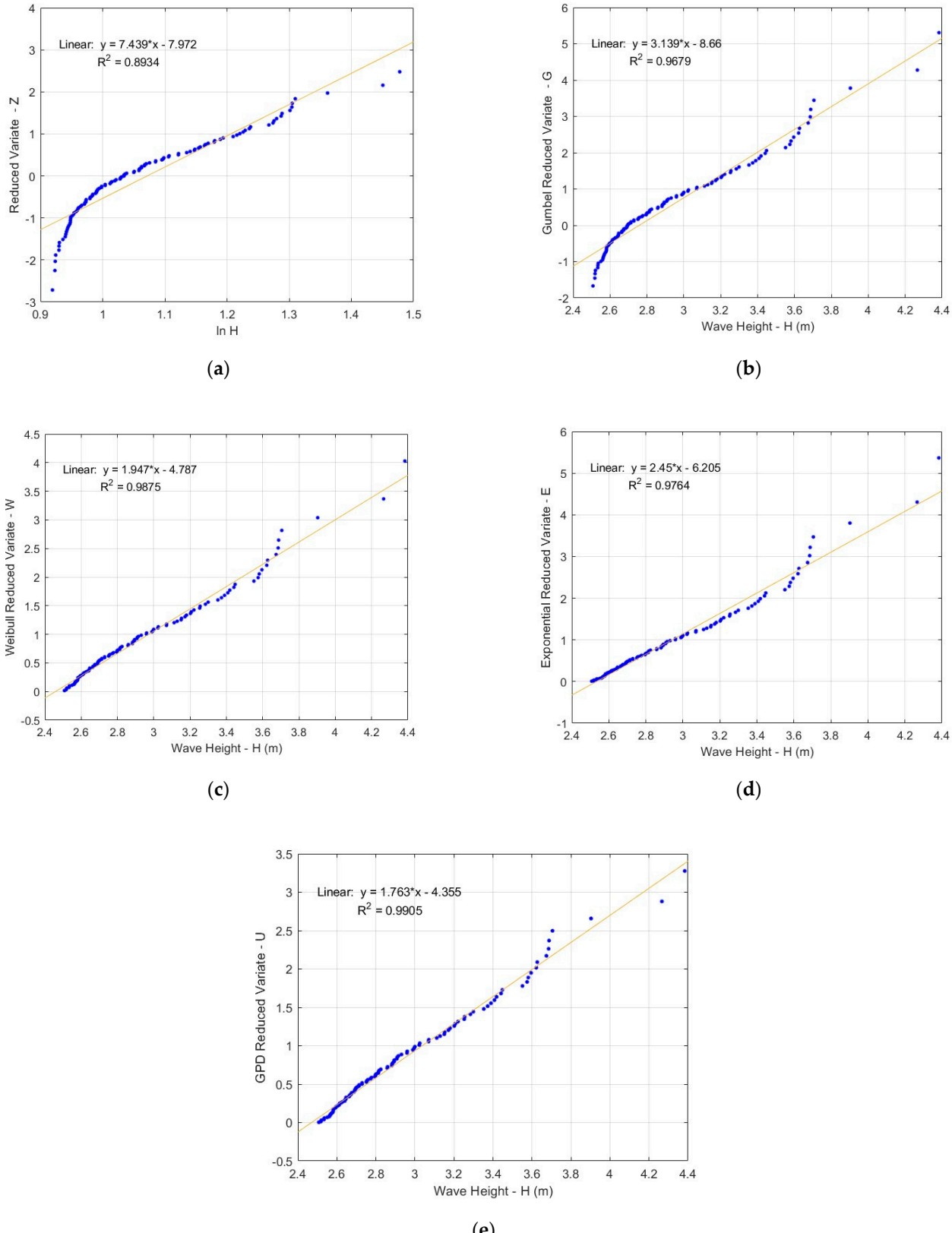

**Figure 4.** Extreme value distribution plots: (**a**) Log-normal distribution; (**b**) Gumbel distribution; (**c**) Weibull distribution; (**d**) Exponential distribution; (**e**) Generalized Pareto Distribution (GPD). $R^2$ sum of squares of the error.

**Table 2.** Sample calculations using four different distributions.

| i | H(m) | LN(H) | Q | P | Z (Log-normal) | G (Gumbel) | W (Weibull) | E (Exponential) | U (GPD) |
|---|------|-------|---|---|----------------|------------|-------------|-----------------|---------|
| 1 | 4.383 | 1.478 | 0.007 | 0.993 | 2.477 | 5.306 | 4.028 | 5.366 | 3.192 |
| 2 | 4.266 | 1.451 | 0.015 | 0.985 | 2.158 | 4.277 | 3.366 | 4.306 | 2.817 |
| 3 | 3.903 | 1.362 | 0.024 | 0.976 | 1.972 | 3.777 | 3.038 | 3.803 | 2.605 |
| 4 | 3.705 | 1.310 | 0.033 | 0.967 | 1.836 | 3.443 | 2.815 | 3.470 | 2.452 |
| 5 | 3.688 | 1.305 | 0.042 | 0.958 | 1.728 | 3.191 | 2.646 | 3.220 | 2.330 |
| 6 | 3.685 | 1.304 | 0.051 | 0.949 | 1.637 | 2.988 | 2.510 | 3.021 | 2.227 |
| 7 | 3.674 | 1.301 | 0.060 | 0.940 | 1.558 | 2.818 | 2.394 | 2.855 | 2.138 |
| 8 | 3.627 | 1.289 | 0.069 | 0.931 | 1.487 | 2.671 | 2.294 | 2.712 | 2.059 |
| 9 | 3.622 | 1.287 | 0.077 | 0.923 | 1.423 | 2.542 | 2.206 | 2.588 | 1.988 |
| 10 | 3.596 | 1.280 | 0.086 | 0.914 | 1.365 | 2.427 | 2.127 | 2.477 | 1.923 |
| 11 | 3.581 | 1.276 | 0.095 | 0.905 | 1.310 | 2.323 | 2.056 | 2.377 | 1.864 |
| 12 | 3.575 | 1.274 | 0.104 | 0.896 | 1.260 | 2.227 | 1.990 | 2.286 | 1.808 |

　　　　Each distribution model could be used to extrapolate and predict the wave heights for different return periods as shown in Table 3. The wave height was predicted for the 1-, 20-, 50-, 100-, and 200-year return periods. The table shows variation in the wave heights as suggested by the different wave height models. For instance, the wave height for the 100-year return period ranges from 4.10 to 4.84 m. Hence, Goda (2010) suggests that the degree of goodness of fit could be used to select the distribution model that fits the data the best [20]. This is done simply through the value of the correlation coefficient (r) between the wave heights of the ordered data and the corresponding reduced variate for each distribution. The model with the correlation coefficient value closest to 1 is judged to be the best-fitting distribution. Table 4 shows the shape, scale, and location ($\alpha$, $\beta$, and $\gamma$) parameters of the best-fitting candidate distribution functions, in addition to the sum of squares of the error ($R^2$) and the correlation coefficient (r) of the distribution functions.

**Table 3.** Wave height predictions (m) for different return periods.

| Distribution | 1 | 20 | 50 | 100 | 200 |
|--------------|---|----|----|-----|-----|
| Log-Normal | 3.06 | 3.81 | 3.98 | 4.10 | 4.22 |
| Gumbel | 3.02 | 4.04 | 4.33 | 4.56 | 4.78 |
| Weibull | 2.99 | 4.10 | 4.41 | 4.63 | 4.85 |
| Exponential | 2.96 | 4.18 | 4.55 | 4.84 | 5.12 |
| GPD | 3.00 | 4.03 | 4.24 | 4.37 | 4.48 |

**Table 4.** Parameter values ($\alpha$: shape parameter; $\beta$: scale parameter; $\gamma$: location parameter) of the best-fitting candidate distribution functions, sum of squares of the error $R^2$, and correlation coefficient r.

| Distribution | $\alpha$ | $\beta$ | $\gamma$ | $R^2$ | r |
|--------------|----------|---------|----------|-------|---|
| Log-Normal | - | 0.13 | 1.07 | 0.8934 | 0.9452 |
| Gumbel | - | 0.32 | 2.76 | 0.9679 | 0.9838 |
| Weibull | 1.2 | 0.51 | 2.46 | 0.9875 | 0.9937 |
| Exponential | - | 0.41 | 2.53 | 0.9764 | 0.9881 |
| GPD | 0.2 | 0.57 | 2.47 | 0.9905 | 0.9953 |

## 4. Discussion

　　　　A long-term wave dataset at a site on the Dubai coastline is used to estimate extreme wave heights at different return periods. A comparison between five different extreme wave distribution methods is used for the analysis, namely, Log-normal, Gumbel, Weibull, Exponential, and GPD. The Peak Over Threshold method is used at 2.5 m to extract 113 storm events from the wave dataset. However, this threshold selection method is subjective. Alternatively, for example, Thompson et al. (2009) present automated threshold selection methods for extreme wave analysis [33]. The authors argue that it is a new,

automated, simple, and computationally inexpensive method for selecting an appropriate threshold for a given dataset. Additionally, the automated threshold selection method has shown that it compares well with the subjective method. After threshold selection, different plots are generated to show the relationship between the wave heights of the 113 peak storms and reduced variates of the corresponding distributions. The decision on the best-fitting distribution among all candidate functions is based on the goodness-of-fit criteria, which consider the correlation coefficient and the sum of squares of the error.

The results of the analysis provide estimations for the significant wave heights at different return periods as shown in Table 3. However, a selection of a distribution that best fits the data might be necessary. The Generalized Pareto Distribution with $\alpha = 0.2$ appears to be the distribution that best fits the storm data, as it has the highest correlation coefficient ($r = 0.9953$) compared to the other distribution models. Therefore, this distribution method is preferred over the others for this dataset. Similarly, the Weibull distribution with $\alpha = 1.2$ relatively has a very good fit to the wave data with $r = 0.9937$. The Weibull and GPD distributions are also favored and suggested by Sulis et al. (2017) in the Gulf of Cagliari (South Sardinia, Italy) [24]. This conclusion could be justified as the Weibull and GPD distributions have an extra parameter ($\alpha$) and, therefore, they are more likely to produce a good fit to a straight line. The value of the shape parameter $\alpha$ is determined by trial and error until the best-fitting straight line is produced. Compared to the GPD model, the extrapolated extreme wave estimates are relatively in agreement with the other models for lower return periods. However, for higher return periods, the other models significantly overestimate or underestimate extrapolated values. These conclusions should not be generalized to all datasets and the best-fitting distribution might not always be the GPD distribution. Therefore, extreme wave analysis should be conducted per dataset in order to obtain accurate predictions of significant wave heights. For instance, the study shows that the Exponential distribution provides the highest predictions of significant wave heights for higher return periods. For example, for the 100-year return period, the Exponential distribution estimate (4.84 m) is around 10% higher compared to the best-fitting distribution (4.37 m).

Climate change and sea-level rise affect significant wave heights [34]. A difference in significant wave height values could be observed by comparing the obtained results in this study with the design wave heights of existing projects. For example, according to Hellebrand et al. (2004), the recorded design wave height for the Palm Jumeirah Island revetment for the 100-year return period was 4 m, which is a close project to the wave data location [35]. This wave height is significantly lower than the significant wave height of 4.37 m that was estimated using the GPD distribution for the 100-year return period. Although the type of extreme wave analysis used for the Palm Jumeirah revetment was not specified, which could certainly be a reason behind the difference in the design wave heights, this study could support the reassessment of the existing revetment at Palm Jumeirah Island. Additionally, the results presented in this paper provide estimates of the significant wave heights, which would be used for the design and upgrade of new and existing coastal structures in Dubai such as man-made islands. However, the used wave hindcast is modeled by the dependence between wind forcings and wave heights. There are other climate change-driven factors that influence wave dynamics. Hence, a multivariate analysis could be performed to better identify the influence of sea-level rise on the significant wave height [36–38].

Further studies could be conducted on the wave hindcast dataset. For instance, other distribution methods could be used in order to obtain more accurate results. A three-parameter distribution that could be adopted is the Generalized Extreme Value distribution (GEV), which has become more popular recently [17]. With this dataset, this method could be used, in which extreme events are grouped into annual intervals and the GEV distribution is fitted into the annual maxima dataset. Additionally, other methods could also be tested to measure the goodness of fit of the storm data such as the Chi-square, Kolmogorov–Smirnov, and Anderson–Darling tests.

## 5. Conclusions

In this study, extreme wave analysis distributions are compared to estimate significant wave heights for different return periods to support engineers with the design, repair, and maintenance of structures on the Dubai coast. A 40-year long-term wave hindcast data set covering 1971–2018 obtained from numerical modeling is used for this study. The Peak Over Threshold method at a threshold of 2.5 m is used for filtering the storms from the raw data. Significant wave heights for different return periods are estimated by fitting some of the most widely used distribution methods such as Log-normal, Gumbel, Weibull, Exponential, and GPD distributions. The different distribution methods were used to predict significant wave heights for different return periods. The GPD distribution with $\alpha = 0.2$ has shown the best degree of fitting to the wave data with the correlation coefficient closest to 1 as compared to the other distribution methods. The obtained significant wave heights show a noticeable difference when compared to the design wave height of the Palm Jumeirah revetment. Hence, this study would help in the reassessment of the existing structure. There are many coastal and offshore projects in the Dubai coastal area, and many others are being planned and designed for the future. Therefore, the results presented in this study are expected to be of great benefit for the design of optimal and safe projects in the area.

**Author Contributions:** Conceptualization, K.E. and S.A.; methodology, K.E.; validation, K.E., S.A. and A.G.Y.; formal analysis, K.E. and A.G.Y.; investigation, K.E. and A.G.Y.; resources, S.A.; data curation, K.E. and S.A., writing—original draft preparation, K.E.; writing—review and editing, K.E., S.A. and A.G.Y., visualization, S.A.; supervision, S.A. All authors have read and agreed to the published version of the manuscript.

**Funding:** This research received no external funding.

**Institutional Review Board Statement:** Not applicable.

**Informed Consent Statement:** Not applicable.

**Data Availability Statement:** The National Centers for Environmental Prediction (NCEP) Climate Forecast System Reanalysis (CFSR) dataset presented in this study is openly available in [Bulletin of the American Meteorological Society] at [https://doi.org/10.1175/2010BAMS3001.1].

**Acknowledgments:** The authors would like to acknowledge Filipe Vieira for providing the hindcast dataset used in this study.

**Conflicts of Interest:** The authors declare no conflict of interest.

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
