# Peer review of "Extreme Wave Analysis for the Dubai Coast"

_hydrology, doi:10.3390/hydrology9080144_

Round 1

Reviewer 1 Report

The manuscript has the potential to advance our understanding of Predicting significant wave heights and its application in the context of water related structure and management. Many of the author's arguments, however, require additional citations. Several examples are provided below:

1. In the introduction line number 36 authors mention 'Climate change is one of the challenges that affect cities, coasts, agriculture, water...' I recommend that the authors provide the following references just after the words 'natural ecosystems': (a) Sarker et al. (2019), Critical Nodes in River Networks, Scientific Reports. https://www.nature.com/articles/s41598-019-47292-4, (b) Chini, Nicolas, et al. "The impact of sea level rise and climate change on inshore wave climate: A case study for East Anglia (UK)." Coastal Engineering 57.11-12 (2010): 973-984.

2. In the introduction line number 41 authors mention '...estimate climate change implications on inshore waves and the occurrence of extreme...' I recommend that the authors provide the following references: Chini, Nicolas, et al. "The impact of sea level rise and climate change on inshore wave climate: A case study for East Anglia (UK)." Coastal Engineering 57.11-12 (2010): 973-984.

3. In the introduction line number 50 authors mention '...wave parameters over a long period of time...' I recommend that the authors provide the following references: Sarker, S.; Sarker, T. Spectral Properties of Water Hammer Wave. Appl. Mech. 2022, 3, 799-814. https://doi.org/10.3390/applmech3030047

4. Figure 1 can be enhanced by adding the UAE map (inset) to the existing image.

5. Figure 2 could be enhanced by adding the theoretical concept of significant wave height to the current image.

6. Figure 4 requires improvement. Python and R are examples of freely obtainable software. This type of software should be utilized by authors to generate figures with a more professional appearance

Author Response

 Hydrology Journal – Manuscript ID: hydrology-1816703

(25/07/2022)

EXTREME WAVE ANALYSIS FOR THE DUBAI COAST

by Khaled Elkersh, Serter Atabay, and Abdullah Gokhan Yilmaz

The authors are grateful to the assistant editor and reviewers for their thorough review and constructive comments.  The Authors have spent some considerable effort in thoroughly revising the paper in the light of these comments, believing that it will be very useful to those who are conducting similar research.  Point-by-point responses to the reviewer comments are provided below. 

Comments by Reviewers:
REVIEWER 1:

 Reviewer Comment:

In the introduction line number 36 authors mention 'Climate change is one of the challenges that affect cities, coasts, agriculture, water...' I recommend that the authors provide the following references just after the words 'natural ecosystems': (a) Sarker et al. (2019), Critical Nodes in River Networks, Scientific Reports. https://www.nature.com/articles/s41598-019-47292-4, (b) Chini, Nicolas, et al. "The impact of sea level rise and climate change on inshore wave climate: A case study for East Anglia (UK)." Coastal Engineering 57.11-12 (2010): 973-984.

Response: 

We would like to thank Reviewer 1 for the constructive comments. Sarker et al. (2019) and Chini et al. (2010) references provided by the reviewer are found useful and cited in the revised manuscript.

Reviewer Comment:

In the introduction line number 41 authors mention '...estimate climate change implications on inshore waves and the occurrence of extreme...' I recommend that the authors provide the following references: Chini, Nicolas, et al. "The impact of sea level rise and climate change on inshore wave climate: A case study for East Anglia (UK)." Coastal Engineering 57.11-12 (2010): 973-984.

Response:

The reference mentioned in this comment was already provided and cited in the original manuscript.

Reviewer Comment:

In the introduction line number 50 authors mention '...wave parameters over a long period of time...' I recommend that the authors provide the following references: Sarker, S.; Sarker, T. Spectral Properties of Water Hammer Wave. Appl. Mech. 2022, 3, 799-814. https://doi.org/10.3390/applmech3030047

Response:

Sarker and Sarker (2022) reference provided by the reviewer is found useful and cited in the revised paper.

Reviewer Comment:

Figure 1 can be enhanced by adding the UAE map (inset) to the existing image.

Response:

Figure 1 is now enhanced by adding the study location (inset) to the Dubai map. The whole UAE map was not used as it is very large compared to the study area. Additionally, Figure 1 now matches the title as the study area is in the Emirate of Dubai.

Reviewer Comment:

Figure 2 could be enhanced by adding the theoretical concept of significant wave height to the current image.

Response:

Figure 2 was recreated using MATLAB. The concept of Figure 2 was enhanced and explained more clearly in the revised manuscript. Additionally, the caption of Figure 2 was edited and improved.

Reviewer Comment:

Figure 4 requires improvement. Python and R are examples of freely obtainable software. This type of software should be utilized by authors to generate figures with a more professional appearance

Response:

Figure 4 was recreated and enhanced using MATLAB software

Reviewer 2 Report

Review for Manuscript hydrology-1816703

Title:

Extreme Wave Analysis for the Dubai Coast

Authors:

Khaled Elkersh , Serter Atabay * , Abdullah Gokhan Yilmaz

General comments

The Manuscript Hydrology-1816703 provides an analysis of the extreme wave climate at a selected point along the Dubai coast and it is within the scope of the Journal "Hydrology".

The analysis presented in the Manuscript is of interest for the readers of the Journal and, in general, for the professionals involved in the design of coastal structures in the Dubai coast. Therefore, the publication of the Manuscript is suggested, provided that some major comments listed in the following, are properly addressed by the Authors.

Authors adopt wave hindcasted data, but they do not mention the origin of the dataset. They have to report the origin to allow any other person to repeat the calculations.

Authors use a well known long term probabilistic analysis of the univariate type. This way, the only contribution from this paper is the publication of the results of a long term probabilistic analysis of univariate type for Hs which can be found in confidential technical reports in Dubai Municipality archives. It neglects the novelty from the use of multivariate long term probabilistic analysis (e.g. for Hs and Ts or Tp; e.g. for Hs and Sea Level) by using copula. At least, Authors must mention all this stuff in the Introduction and oropose a future work with the novel multivariate methodologies.

Specific comments

Section 1:

Please provide a bathymetry of the area and cite in the text the depth of the data point you selected.

Section 2:

Please provide a reference to the wave dataset you used: which model has been used to produce it, if it comes from NOAA, etc.

Specify if wave height (line 85) is the significant wave height; if time period is peak wave period of mean wave period.

How the final results are changed if waves coming from all the directions are considered (not only waves coming from northwest)?

Can the Authors specify at which quantile of the distribution of the significant wave height, the value 2.5 m (the threshold you adopted) corresponds to?

Line 148 "Sulis et al. (2017) suggest that the Generalized Pareto Distribution (GPD) shown in Equation 11..." this sentence is placed before deriving the actual Eq. 11 so the readers need to jump few lines ahead in the text to get to Eq. 11 and that is not convenient. Please rearrange this part.

There are two Eq. 11 in the manuscript; please fix the typo.

Section 4:

Please provide the following reference to the statement at line 248 "Climate change and sea level rise affect the significant wave heights ":

Tomasicchio, G.R., Salvadori, G., Lusito, L. et al. (2021). A Statistical Analysis of the Occurrences of Critical Waves and Water Levels for the Management of the Operativity of the MoSE System in the Venice Lagoon. Stoch Environ Res Risk Assess. https://doi.org/10.1007/s00477-021-02133-7

The statement at line 256: "Therefore, extreme wave analysis should be conducted for the Dubai coast in order to maintain or upgrade the existing coastal structures" as a tool to mitigate climate change effects is not very precise for the following reason.

Extreme wave analysis, consisting in analyzing a long dataset of wave hindcasts (as it is done in the Manuscript, by considering wave hindcasts from 1979 and 2018) cannot represent precisely the effect of climate change, because 1) the record is not long enough to show the effects of climate change 2) it is not clear (because the Authors do not provide enough details) if the dataset incorporates data assimilation techniques, i.e. whether it keeps into account observations of possibly higher significant wave heights recorded in more recent years  3) as the Authors said, there are other climate-change driven factors that influence the wave dynamics and multivariate analysis should be performed in order to establish the influence of sea level rise. See for example:

Ferreira, J.A., Guedes Soares, C., 2002. Modelling bivariate distributions of significant wave height and mean wave period. Appl. Ocean Res. 24, 31–45. http://dx.doi.org/10.1016/S0141-1187(02)00006-8.

Genest, C., Favre, A., 2007. Everything you always wanted to know about copula modeling but were afraid to ask. J. Hydrol. Eng. 12, 347–368.

De Michele, C., Salvadori, G., Passoni, G., Vezzoli, R., 2007. A multivariate model of sea storms using copulas. Coast. Eng. 54, 734–751.

There are many other more recent references for the same Authors.

Please rectify the statement and add the references to the bibliography.

Author Response

EXPLANATION OF REVISIONS

 Hydrology Journal – Manuscript ID: hydrology-1816703

(25/07/2022)

  EXTREME WAVE ANALYSIS FOR THE DUBAI COAST

by Khaled Elkersh, Serter Atabay, and Abdullah Gokhan Yilmaz

The authors are grateful to the assistant editor and reviewers for their thorough review and constructive comments.  The Authors have spent some considerable effort in thoroughly revising the paper in the light of these comments, believing that it will be very useful to those who are conducting similar research.  Point-by-point responses to the reviewer comments are provided below. 

Comments by Reviewers:

REVIEWER 2:

Reviewer Comment:

Please provide a bathymetry of the area and cite in the text the depth of the data point you selected.

Response:

We would like to thank Reviewer 2 for the constructive comments. A description of the average depth of the UAE territorial water is now provided in the manuscript. Additionally, the depth of the selected data point is also provided in the revised paper.

Reviewer Comment:

Please provide a reference to the wave dataset you used: which model has been used to produce it, if it comes from NOAA, etc.

Response:

The wave hindcast used in the paper was referenced in the revised paper and the model used to generate it was also described.

Reviewer Comment:

Specify if wave height (line 85) is the significant wave height; if time period is peak wave period of mean wave period.

Response:

The statement in line 85 is fixed and specified in the revised paper.

Reviewer Comment:

How the final results are changed if waves coming from all the directions are considered (not only waves coming from northwest)?

Response:

This comment was addressed in the revised manuscript. Storms coming from the northwest direction are very critical and often called Shamals. Minor or no changes would be observed in the results if all directions are considered as the number of isolated storms at the 2.5 m threshold will still remain 113 storms. Considering only the northwest direction was properly justified in the revised manuscript.

Reviewer Comment:

Can the Authors specify at which quantile of the distribution of the significant wave height, the value 2.5 m (the threshold you adopted) corresponds to?

Response:

This comment was addressed in the revised manuscript by mentioning the quartile of the distribution that the 2.5 m threshold corresponds to.

Reviewer Comment:

Line 148 "Sulis et al. (2017) suggest that the Generalized Pareto Distribution (GPD) shown in Equation 11..." this sentence is placed before deriving the actual Eq. 11 so the readers need to jump few lines ahead in the text to get to Eq. 11 and that is not convenient. Please rearrange this part.

Response:

This comment was addressed in the revised paper by editing and rearranging the statement in line 148.

Reviewer Comment:

There are two Eq. 11 in the manuscript; please fix the typo.

Response:

The typo was fixed in the revised paper

Reviewer Comment:

Please provide the following reference to the statement at line 248 "Climate change and sea level rise affect the significant wave heights ":

Tomasicchio, G.R., Salvadori, G., Lusito, L. et al. (2021). A Statistical Analysis of the Occurrences of Critical Waves and Water Levels for the Management of the Operativity of the MoSE System in the Venice Lagoon. Stoch Environ Res Risk Assess. https://doi.org/10.1007/s00477-021-02133-7

Response:

The reference was added to the statement in line 248

Reviewer Comment:

The statement at line 256: "Therefore, extreme wave analysis should be conducted for the Dubai coast in order to maintain or upgrade the existing coastal structures" as a tool to mitigate climate change effects is not very precise for the following reason.

Extreme wave analysis, consisting in analyzing a long dataset of wave hindcasts (as it is done in the Manuscript, by considering wave hindcasts from 1979 and 2018) cannot represent precisely the effect of climate change, because 1) the record is not long enough to show the effects of climate change 2) it is not clear (because the Authors do not provide enough details) if the dataset incorporates data assimilation techniques, i.e. whether it keeps into account observations of possibly higher significant wave heights recorded in more recent years  3) as the Authors said, there are other climate-change driven factors that influence the wave dynamics and multivariate analysis should be performed in order to establish the influence of sea level rise. See for example:

Ferreira, J.A., Guedes Soares, C., 2002. Modelling bivariate distributions of significant wave height and mean wave period. Appl. Ocean Res. 24, 31–45. http://dx.doi.org/10.1016/S0141-1187(02)00006-8.

Genest, C., Favre, A., 2007. Everything you always wanted to know about copula modeling but were afraid to ask. J. Hydrol. Eng. 12, 347–368.

De Michele, C., Salvadori, G., Passoni, G., Vezzoli, R., 2007. A multivariate model of sea storms using copulas. Coast. Eng. 54, 734–751.

There are many other more recent references for the same Authors.

Please rectify the statement and add the references to the bibliography.

Response:

The comment provided by the reviewer was very useful. Therefore, the novel on multivariate methodologies was addressed in the discussion section of the revised manuscript by referring to any future work that could be done in this area in the future. Additionally, the statement was clarified and re-explained, and the references provided by the reviewer were also used and cited in the revised manuscript.

Reviewer 3 Report

Article reference: hydrology-1816703

Title: Extreme Wave Analysis for the Dubai Coast

Author(s): Khaled Elkersh, Serter Atabay and Abdullah Gokhan Yilmaz

The authors present the result of commonly used extreme wave analysis distribution methods applied to a long-term wave hindcast at a point in the Arabian Gulf near the coastline of Dubai, United Arab Emirates for a total period of 40 years, starting from 1st of January, 1979 to 31st of December, 2018. The authors show the effect of climate change and sea-level rise on the design wave height.

Results look interesting and useful in the context of wave dynamics and to explain the characteristics of the sea area and also they will be helpful in effectively designing the coastal projects that are planned near the Dubai coastline.  I recommend that the paper be included in the published journal. However, I would suggest a number of alterations which I believe necessary to improve the readability of the work.

1. The authors should complement the literature on Extreme Wave Analysis: the shape parameter, higher moments and variance

2. Why you didn't use other parameters than the squares of the error (R²) and the correlation coefficient (r), such as the variance or other parameter

3. The quality of all Figure should be improved

4. In discussion and conclusion, the results should be discussed in comparison to the previous studies and the authors should justify in more detail the novelty of their results.

Author Response

EXPLANATION OF REVISIONS

 Hydrology Journal – Manuscript ID: hydrology-1816703 (25/07/2022)

 EXTREME WAVE ANALYSIS FOR THE DUBAI COAST

by Khaled Elkersh, Serter Atabay, and Abdullah Gokhan Yilmaz

The authors are grateful to the assistant editor and reviewers for their thorough review and constructive comments.  The Authors have spent some considerable effort in thoroughly revising the paper in the light of these comments, believing that it will be very useful to those who are conducting similar research.  Point-by-point responses to the reviewer comments are provided below. 

Comments by Reviewers:

REVIEWER 3:

Reviewer Comment:

The authors should complement the literature on Extreme Wave Analysis: the shape parameter, higher moments and variance

Response: 

We would like to thank Reviewer 3 for the constructive comments. This comment was addressed in multiple areas in the revised manuscript. A few extra sentences were added to the literature on Extreme Wave Analysis and the shape parameter.

Reviewer Comment:

Why you didn't use other parameters than the squares of the error (R²) and the correlation coefficient (r), such as the variance or other parameter.

Response: 

The adopted method of using the squares of the error (R²) and the correlation coefficient (r) is a commonly used and accepted way of comparing the different used distribution models and is enough in identifying the best fitting model for our results.

Reviewer Comment:

The quality of all Figure should be improved

Response:

All figures were recreated, and the quality of the figures was improved in the revised paper

Reviewer Comment:

In discussion and conclusion, the results should be discussed in comparison to the previous studies and the authors should justify in more detail the novelty of their results.

Response:

This comment was addressed in the revised manuscript. The results were explained and compared with previous studies. For instance, the obtained results were compared with a similar case study in the Gulf of Cagliari (South Sardinia, Italy). Additionally, the results were also compared with the design wave height used for the 100-year return period of The Palm Jumeirah Island. The difference between the obtained results and the used design wave height highlights the novelty of the study and the need for extreme wave analysis by professionals and for the design and upgrade of coastal structures. These details were explained in the discussion and conclusion of the revised manuscript.

Round 2

Reviewer 1 Report

Thanks for the revision. The manuscript improved significantly. 

Author Response

Thanks for your comments.

Reviewer 2 Report

Review for Manuscript hydrology-1816703

 Title:

Extreme Wave Analysis for the Dubai Coast

 Authors:

Khaled Elkersh , Serter Atabay * , Abdullah Gokhan Yilmaz

Authors have partially fulfilled the requests from this reviewer. As a major concern, they did not pay attention to the following comment:

Extreme wave analysis, consisting in analyzing a long dataset of wave hindcasts (as it is done in the Manuscript, by considering wave hindcasts from 1979 and 2018) cannot represent precisely the effect of climate change, because 1) the record is not long enough to show the effects of climate change 2) it is not clear (because the Authors do not provide enough details) if the dataset incorporates data assimilation techniques, i.e. whether it keeps into account observations of possibly higher significant wave heights recorded in more recent years  3) as the Authors said, there are other climate-change driven factors that influence the wave dynamics and multivariate analysis should be performed in order to establish the influence of sea level rise. “

As a consequence, sentences like "Additionally, this study shows the effect of climate change and sea-level rise on the design wave height." must be removed from the entire manuscript. Climate change is still present in the Keywords.

With this in mind, all the manuscript has to be rehandled cause 40 ys of observations are sufficiently enough for a serious extreme analysis, but not for a study on the influence of climate change on Meteomarine conditions.

Another issue is that the manuscript still does not provide a reference to the wave dataset used: which model has been used to produce it, if it comes from NOAA, etc. ?

Authors attempt to justify the influence of climate change with this sentence: “For example, according to Hellebrand et al. (2004), the recorded design wave height for the Palm Jumeirah Island revetment for the 100-year return period was 4 m,”.  But Hellebrand et al (2004) simply stated “Because of shoaling effects, the wave height decreases to approximately 4.0 m at the outer point of the breakwater” and did not refer about the type of extreme analysis they made which can certainly be the reason for the difference between the results for the 100 yrs return period.

Author Response

Reviewer Comment:

As a consequence, sentences like "Additionally, this study shows the effect of climate change and sea-level rise on the design wave height." must be removed from the entire manuscript. Climate change is still present in the Keywords.

Response:

This comment was addressed in the manuscript. Statements were removed from the abstract, discussion, and conclusion as suggested by the reviewer.

Reviewer Comment:

With this in mind, all the manuscript has to be rehandled cause 40 ys of observations are sufficiently enough for a serious extreme analysis, but not for a study on the influence of climate change on Meteomarine conditions.

Response:

The manuscript was rehandled to focus mainly on extreme wave analysis rather than the effect of climate change on Meteomarine conditions.

Reviewer Comment:

Another issue is that the manuscript still does not provide a reference to the wave dataset used: which model has been used to produce it, if it comes from NOAA, etc. ?

Response:

This comment was addressed, and a reference was provided in the revised manuscript. Additionally, the data availability statement was modified accordingly.

Reviewer Comment:

Authors attempt to justify the influence of climate change with this sentence: “For example, according to Hellebrand et al. (2004), the recorded design wave height for the Palm Jumeirah Island revetment for the 100-year return period was 4 m,”.  But Hellebrand et al (2004) simply stated “Because of shoaling effects, the wave height decreases to approximately 4.0 m at the outer point of the breakwater” and did not refer about the type of extreme analysis they made which can certainly be the reason for the difference between the results for the 100 yrs return period.

Response:

This comment was addressed in the revised manuscript and the statement was fixed and rephrased accordingly. Although this comment is certainly valid, the results presented in the study could be useful in the reassessment or upgrade of the Palm Jumeirah revetment.